# Partitioning to ordered membrane domains regulates the kinetics of secretory traffic

Ivan Castello-Serrano[1], Frederick A Heberle[2], Barbara Diaz-Rohrer[3], Rossana Ippolito[1], Carolyn R Shurer[1], Pablo Lujan[4], Felix Campelo[4], Kandice R Levental[1]*, Ilya Levental[1]*

[1]Department of Molecular Physiology and Biological Physics, Center for Membrane and Cell Physiology, University of Virginia, Charlottesville, United States; [2]Department of Chemistry, The University of Tennessee, Knoxville, United States; [3]Broad Institute of MIT and Harvard, Boston, United States; [4]ICFO-Institut de Ciencies Fotoniques, The Barcelona Institute of Science and Technology, Barcelona, Spain

**\*For correspondence:**
krl6c@virginia.edu (KRL);
il2sy@virginia.edu (IL)

**Abstract** The organelles of eukaryotic cells maintain distinct protein and lipid compositions required for their specific functions. The mechanisms by which many of these components are sorted to their specific locations remain unknown. While some motifs mediating subcellular protein localization have been identified, many membrane proteins and most membrane lipids lack known sorting determinants. A putative mechanism for sorting of membrane components is based on membrane domains known as lipid rafts, which are laterally segregated nanoscopic assemblies of specific lipids and proteins. To assess the role of such domains in the secretory pathway, we applied a robust tool for synchronized secretory protein traffic (RUSH, *R*etention *U*sing *S*elective *H*ooks) to protein constructs with defined affinity for raft phases. These constructs consist solely of single-pass transmembrane domains (TMDs) and, lacking other sorting determinants, constitute probes for membrane domain-mediated trafficking. We find that while raft affinity can be sufficient for steady-state PM localization, it is not sufficient for rapid exit from the endoplasmic reticulum (ER), which is instead mediated by a short cytosolic peptide motif. In contrast, we find that Golgi exit kinetics are highly dependent on raft affinity, with raft preferring probes exiting the Golgi ~2.5-fold faster than probes with minimal raft affinity. We rationalize these observations with a kinetic model of secretory trafficking, wherein Golgi export can be facilitated by protein association with raft domains. These observations support a role for raft-like membrane domains in the secretory pathway and establish an experimental paradigm for dissecting its underlying machinery.

## eLife assessment

In this **important** study, Castello-Serrano and colleagues describe, model and quantify the role of transmembrane domains in protein sorting in the secretory pathway, first at the ER and subsequently at the Golgi. **Convincing** data support the role of a cytoplasmic motif in ER exit, while further experiments are necessary to support a direct connection between the phase partitioning capability of the transmembrane regions and the sorting potential of domains at the Golgi/TGN.

## Introduction

Secretory and membrane proteins are synthesized in the endoplasmic reticulum (ER) followed by export of non-ER-resident proteins to the Golgi prior to sorting to their ultimate cellular location (*Lippincott-Schwartz et al., 2000*). Soluble and transmembrane proteins (TMPs) require somewhat

different mechanisms to exit the ER, but both are dependent on the coat protein complex II (COPII), a multi-protein assembly recruited to the ER membrane to sort cargo and facilitate membrane deformation (*Otte and Barlowe, 2004*; *Sato and Nakano, 2007*). Cargo proteins are typically recognized for ER exit by binding to COPII subunits via one of several short peptide motifs on their cytoplasmic tails (*Barlowe, 2003b*; *Kuehn et al., 1998*). Proteins that lack these motifs can be sorted by binding to motif-containing adaptors (*Castillon et al., 2011*; *di Ronza et al., 2018*). COPII mediated ER-to-Golgi transport is believed to proceed via vesicles (*Gomez-Navarro et al., 2020*; *McCaughey and Stephens, 2019*); however, recent observations suggest COPII mediates constrictions in continuous membrane structures connecting ER and Golgi membranes (*Weigel et al., 2021*).

The mechanisms and determinants of secretory and membrane protein sorting through the non-contiguous cisternae of the Golgi apparatus are less well understood. Non-resident proteins of the Golgi flux through distinct sub-compartments from cis-to-trans Golgi via several possible non-exclusive paths, including vesicles that exchange between static cisternae and/or dynamic cisternae that themselves change their composition and function over time (*Lujan and Campelo, 2021*; *Pantazopoulou and Glick, 2019*). Post-Golgi sorting generally occurs at the trans-Golgi network (TGN), where exiting proteins are packaged into transport carriers targeted to other organelles (*Luini et al., 2008*; *Lujan et al., 2022*; *Nishimura et al., 2002*; *Tan and Gleeson, 2019*). These sorting steps are at least partially mediated by clathrin (*Ford et al., 2021*) and its cargo-binding adapter proteins (*Farías et al., 2012*; *Nishimura et al., 2002*; *Park and Guo, 2014*). In some cases (e.g. mannose-6-phosphate), the principles of sorting are well understood (*Puertollano et al., 2001*). But for many other TMPs, the specific determinants of their subcellular trafficking itineraries and ultimate steady-state location are unknown. Even more mysterious are the rules for lipid sorting between various membranes, though lipid transfer machineries at membrane contact sites are likely important (*Elbaz and Schuldiner, 2011*; *Mesmin et al., 2013*; *Phillips and Voeltz, 2016*).

In parallel with polymerizing coats and motif-recognizing adapters, a putative scheme for sorting membrane lipids and proteins relies on nanodomains known as lipid rafts (*Levental et al., 2020*; *Simons and Ikonen, 1997*). Rafts arise due to the intrinsic capacity of biomembranes to laterally separate into coexisting ordered and disordered domains (*Elson et al., 2010*). AltThough still widely debated, recently accumulating evidence has validated many of the key predictions of the lipid raft hypothesis (*Levental et al., 2020*; *Sezgin et al., 2017*). One such predictions, indeed the original function for which lipid rafts were proposed, is that rafts facilitate polarized sorting in epithelial cells, assembling lipids and proteins for delivery from the trans-Golgi network to the apical PM (*Lafont et al., 1999*; *Simons and Ikonen, 1997*). Rafts have also been implicated in endocytic sorting (*Gagescu et al., 2000*), with some proteins relying on raft affinity for recycling to the PM after endocytosis (*Diaz-Rohrer et al., 2023*; *Diaz-Rohrer et al., 2014b*).

Key evidence supporting the capacity for biomembranes to form raft domains is provided by studies of plasma membranes isolated from living cells as Giant Plasma Membrane Vesicles (GPMVs). Such GPMVs separate into coexisting ordered and disordered phases that laterally sort membrane components according to their preference for certain membrane environments (*Levental and Levental, 2015a*). Namely, saturated lipids, glycolipids, GPI-anchored proteins, and selected TMPs co-enrich within a relatively tightly packed lipid phase (termed the 'raft phase'; *Sezgin et al., 2012*), away from unsaturated phospholipids and most TMPs (*Castello-Serrano et al., 2020*; *Levental et al., 2011*). GPMVs thus provide a robust tool to quantitatively assess the intrinsic affinity of membrane components for raft-like domains in biomembranes. Importantly, the preference of some TMPs for raft domains is tightly correlated with their subcellular localization, with raft association being necessary and sufficient for PM localization (*Diaz-Rohrer et al., 2014b*). For these, loss of raft affinity leads to accumulation in endo-lysosomes and ultimate degradation (*Diaz-Rohrer et al., 2023*).

While these observations, among many others (*Abrami et al., 2003*; *Diaz-Rohrer et al., 2014a*; *Fabbri et al., 2005*; *Glebov et al., 2006*; *Sabharanjak et al., 2002*), suggest that lipid-driven microdomains are involved in endocytosis and recycling, the role of rafts in the secretory pathway remains controversial and poorly understood due in large part to methodological limitations. Classical protocols for measuring raft association have been artifact-prone, non-quantitative, and difficult to interpret, while experiments relying on raft disruption are often pleiotropic (*Levental et al., 2020*; *Munro, 2003*). To circumvent these issues and directly interrogate the role of raft association in secretory traffic, we constructed a panel of minimal TMP probes with defined preferences for raft domains and

measured their post-ER trafficking itineraries using RUSH, a robust tool for synchronized secretory traffic (*Boncompain et al., 2012*). We observed that raft association is sufficient for the steady-state distribution of some proteins, ER exit kinetics are largely determined by a sorting motif that likely mediates cargo association with COPII machinery. In contrast, raft affinity is sufficient to confer rapid efflux from the Golgi. We rationalize these observations with a kinetic model that generates a quantitative description of temporal transport of secretory cargo as a function of its association with various membrane subdomains. This model reproduces experimental observations and predicts separation of Golgi membranes into coexisting domains. Consistently, we microscopically observe separation of raft from non-raft probes in Golgi compartments and disruption of raft-associated Golgi efflux by inhibition of raft lipid synthesis. These observations reveal a role for lipid-driven domains in Golgi cargo sorting and provide a quantitative description of membrane protein dynamics through the secretory pathway.

## Results
### ER exit rates are determined more by cytosolic sorting motifs than raft affinity

In previous work, we used protein constructs comprised solely of transmembrane domains (TMDs) to interrogate raft-dependent recycling in the endocytic system (*Diaz-Rohrer et al., 2023*; *Diaz-Rohrer et al., 2014b*; *Lorent et al., 2017*). The preference of these probes for raft domains can be quantified directly by measuring their partitioning between ordered (raft) and disordered (non-raft) phases of isolated GPMVs (*Levental and Levental, 2015b*; *Sezgin et al., 2012*), as shown in *Figure 1A–B*. As previously shown (*Diaz-Rohrer et al., 2014b*; *Lorent et al., 2017*), the TMD of a single-pass transmembrane adapter called Linker for Activation of T-cells (LAT) preferentially partitions into the raft phase of GPMVs, away from the non-raft phase lipid marker (F-DiO; *Figure 1A*, left). A model raft-excluded TMD is a 22-Leu construct (i.e. allL) which strongly prefers the non-raft phase (*Figure 1A*, right). This behavior can be quantified via the raft partition coefficient ($K_{p,raft}$) defined as the ratio of background-subtracted intensities for various constructs in the raft versus non-raft phases (*Figure 1B*). The LAT TMD (open symbols) retains the raft affinity of full-length LAT, while allL-TMD, either on its own or inserted into full-length LAT (LAT-allL), has very low raft affinity (*Figure 1B*).

To quantitatively assay the effects of raft affinity on the kinetics of secretory trafficking, we inserted these probes into the RUSH system, which allows synchronized release and tracking through the secretory system (*Figure 1C*). This assay is based on the reversible interaction of a target protein (fused to streptavidin-binding peptide, SBP) with a selective 'hook'; for example, streptavidin stably anchored in the ER via a KDEL motif. The strong interaction between the hook and SBP retains the protein of interest (POI) in the ER. Introduction of biotin causes rapid release of the POI, which is then tracked via a fluorescent tag. SBP-tagged versions (i.e. RUSH) of all constructs assayed here had indistinguishable raft affinity from non-SBP versions (RFP-only) (*Figure 1B*).

The isolated TMD of LAT was sufficient to recapitulate the steady-state plasma membrane (PM) localization of its full-length protein (*Figure 1—figure supplement 1*), consistent with previous reports (*Diaz-Rohrer et al., 2014b*; *Lorent et al., 2017*). In clear contrast, LAT-TMD did not recapitulate the ER exit kinetics of full-length LAT (*Figure 1D*). 45 mins after biotin addition, LAT had completely exited the ER and was localized almost exclusively to a bright perinuclear region, likely the Golgi. After 90 min, most full-length LAT achieved its steady-state localization at the PM. In contrast, LAT-TMD expressed in the same cell was still in the ER after 90 min and clearly accumulated at the PM only after several hours. Such slow kinetics were a challenge for live-cell imaging, so ER exit was quantified on a population level, by fixing cells at various time points after biotin addition and quantifying the fraction of cells with observable ER localization. The temporal reduction in the fraction of ER-positive cells could be reasonably approximated by a single-exponential fit for all constructs (e.g. *Figure 1E*), revealing that the half-time for ER exit was >fourfold faster for LAT (0.7 hr) than for LAT-TMD (3.1 hr; *Figure 1E–G*).

We next tested whether raft affinity affected ER exit kinetics. To this end, we generated a RUSH construct of full-length LAT whose TMD was replaced with one that has minimal raft affinity (LAT-allL, *Figure 1A*). The general kinetics of ER exit of this non-raft LAT were comparable to LAT, with clear Golgi accumulation after 60 min of biotin. Post-Golgi accumulation was prominent at 90 min for

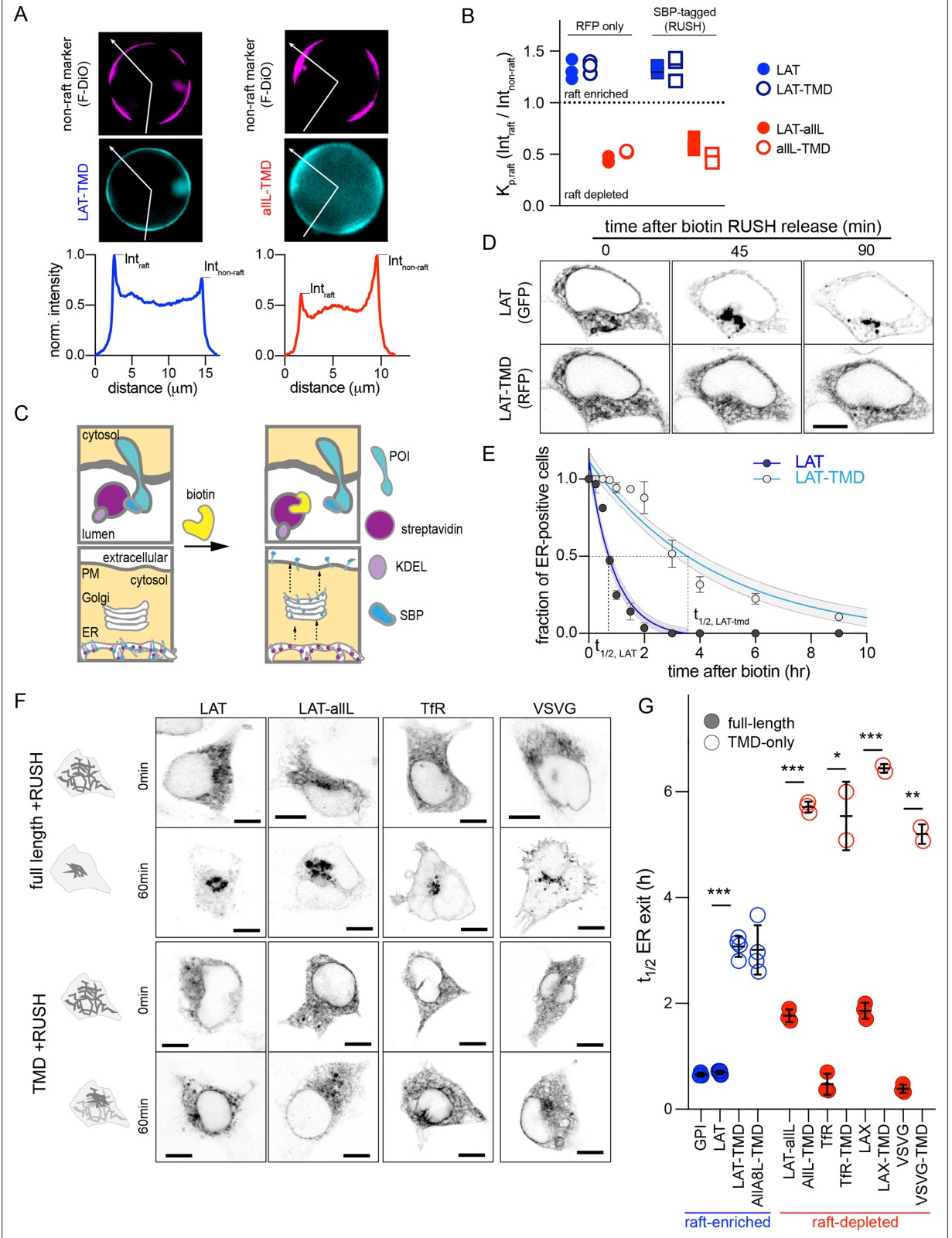

**Figure 1.** Full-length proteins exit the ER faster than TMD-only versions regardless of raft affinity. (**A**) Exemplary images of GPMVs from cells expressing raft-preferring LAT-TMD (left) or non-raft-preferring allL-TMD (right). The RFP-tagged TMDs are shown in cyan; magenta shows the disordered phase marker Fast DiO (F-DiO). Bottom row shows fluorescence intensity line scans along white lines shown in cyan images, revealing protein partitioning between raft and non-raft phases. (**B**) The ratio of intensities in raft versus non-raft phase is the raft partition coefficient ($K_{p,raft}$). LAT-TMD and full-length

*Figure 1 continued on next page*

*Figure 1 continued*

LAT are enriched in the raft phase while allL-TMD (and full-length LAT with allL-TMD, LAT-allL) are largely depleted from raft phase. SBP-tagging (for RUSH assay) has no effect on raft affinity. Symbols represent 3 independent experiments with >10 GPMVs/experiment. All blue labeled constructs are not statistically different from one another, and each is $P<0.01$ different from all red constructs. (**C**) Schematic of RUSH assay. (**D**) Confocal images of co-transfected LAT-EGFP and LAT-TMD-mRFP at various time points after biotin introduction. Full-length LAT exits ER faster. (**E**) Fraction of ER-positive cells decreases over time, allowing quantitative estimation of ER exit kinetics ($t_{1/2}$). Symbols represent average +/-st.dev. from >3 independent experiments. Fits represent exponential decays with shading representing 95% confidence intervals. (**F**) Confocal images of various full-length and TMD-only RUSH constructs (RFP-tagged) at 0 and 60 min after biotin introduction. (**G**) Quantification of $t_{1/2}$ for ER exit comparing full-length and TMD-only proteins (blue represents raft-enriched proteins, red = raft-depleted; see ***Supplementary file 1*** for $K_{p,raft}$ quantifications). Bars represent average ± st.dev. from three independent experiments; *p<0.05, **p<0.01, ***p<0.001. All scale bars correspond to 5 µm. Original data quantification can be found in the Source Data files.

The online version of this article includes the following source data and figure supplement(s) for figure 1:

**Source data 1.** Exemplary images of GPMVs from cells expressing raft-preferring or non-raft-preferring TMDs.

**Source data 2.** Raft partitioning coefficient of TMDs with and without the SBP-tag.

**Source data 3.** Confocal images of co-transfected full-length and TMD-only versions of LAT protein at various time points after biotin introduction.

**Source data 4.** Quantification of time of residency for ER exit of full-length and TMD-only proteins of raft-enriched and raft-depleted proteins.

**Figure supplement 1.** Representative images of RUSH constructs after overnight (>10 hr) treatment with biotin.

both LAT and LAT-allL (the steady state localization of LAT-allL is in endolysosomes, *Figure 1—figure supplement 1*, as previously explained *Diaz-Rohrer et al., 2023*; *Diaz-Rohrer et al., 2014b*). In contrast, the isolated allL-TMD had much slower ER exit kinetics (*Figure 1F–G*). This trend was generalizable to several other proteins, with full-length versions having >threefold faster ER exit kinetics than the TMD-only versions, regardless of their raft affinity (*Figure 1F–G*, *Supplementary file 1*). It is worth noting that isolated TMDs still exited the ER and reached a steady-state localization after several hours (*Figure 1—figure supplement 1*), despite no direct interactions with cytosolic trafficking machinery (e.g. COPII or clathrin). Altogether, we conclude that features present in cytosolic domains play a dominant role over TMD-determined raft affinity in ER exit.

## LAT has a C-terminal ΦxΦxΦ ER exit motif

LAT is comprised of a minimal N-terminal ectodomain (<5 residues), a single-pass TMD, and a largely disordered cytosolic domain (CTD). Since LAT-TMD showed slow ER exit, we inferred that the determinant of rapid ER exit is likely located in the CTD and created a series of C-terminal truncations to locate the signal (*Figure 2A*). The two smaller C-term truncations (ΔCt1 and ΔCt2) had no effect on ER exit kinetics, behaving like full-length LAT (*Figure 2B–C*). In contrast, both larger truncations (ΔCt3 and ΔCt4) were significantly slower, behaving like LAT-TMD (*Figure 2B–C*). Thus, the motif facilitating fast ER exit of LAT is located within residues 140–185 (*Figure 2A*). We analyzed this fragment for possible COPII binding motifs (*Barlowe, 2003a*; *Mikros and Diallinas, 2019*) and identified [146]AAPSA[150], which corresponds to a ΦxΦxΦ motif (Φ=hydrophobic residue, x=spacer) (*Otsu et al., 2013*; *Figure 2A-inset*). Point mutations in this putative motif confirmed that P148 and A150 were essential for fast ER exit, while other neighboring residues were not (*Figure 2D–E* and *Figure 2—figure supplement 1A*). We note that both these residues are nearly completely conserved in LAT from 30 species (*Figure 2A* and *Figure 2—figure supplement 1B*). Finally, inserting this AaPsA motif into LAT-TMD significantly accelerated its ER exit, nearly recapitulating that of full-length LAT (*Figure 2F*). Thus, we conclude that fast ER exit of LAT is mediated by a ΦxΦxΦ motif in the cytosolic CTD, which likely mediates COPII association.

## Raft affinity determines Golgi exit kinetics of LAT

The RUSH system can be adapted to measure exit rates from Golgi by changing the 'hook' to Golgin84, a resident protein of cis-Golgi (*Diao et al., 2003*). As expected, all RUSH constructs (LAT, LAT-TMD, LAT-allL, allL-TMD) co-transfected with this hook were localized almost exclusively in the Golgi without biotin (*Figure 3A*). Within 90 min of biotin introduction, LAT exited the Golgi, achieving the expected steady-state PM localization (*Figure 3A*). LAT-allL was clearly slower, with abundant Golgi signal remaining after 90 min. Tracking the kinetics by quantitative imaging, LAT exited the Golgi ~2.5-fold faster than LAT-allL (*Figure 3B–C*). Strikingly, the TMD-only versions of these constructs behaved

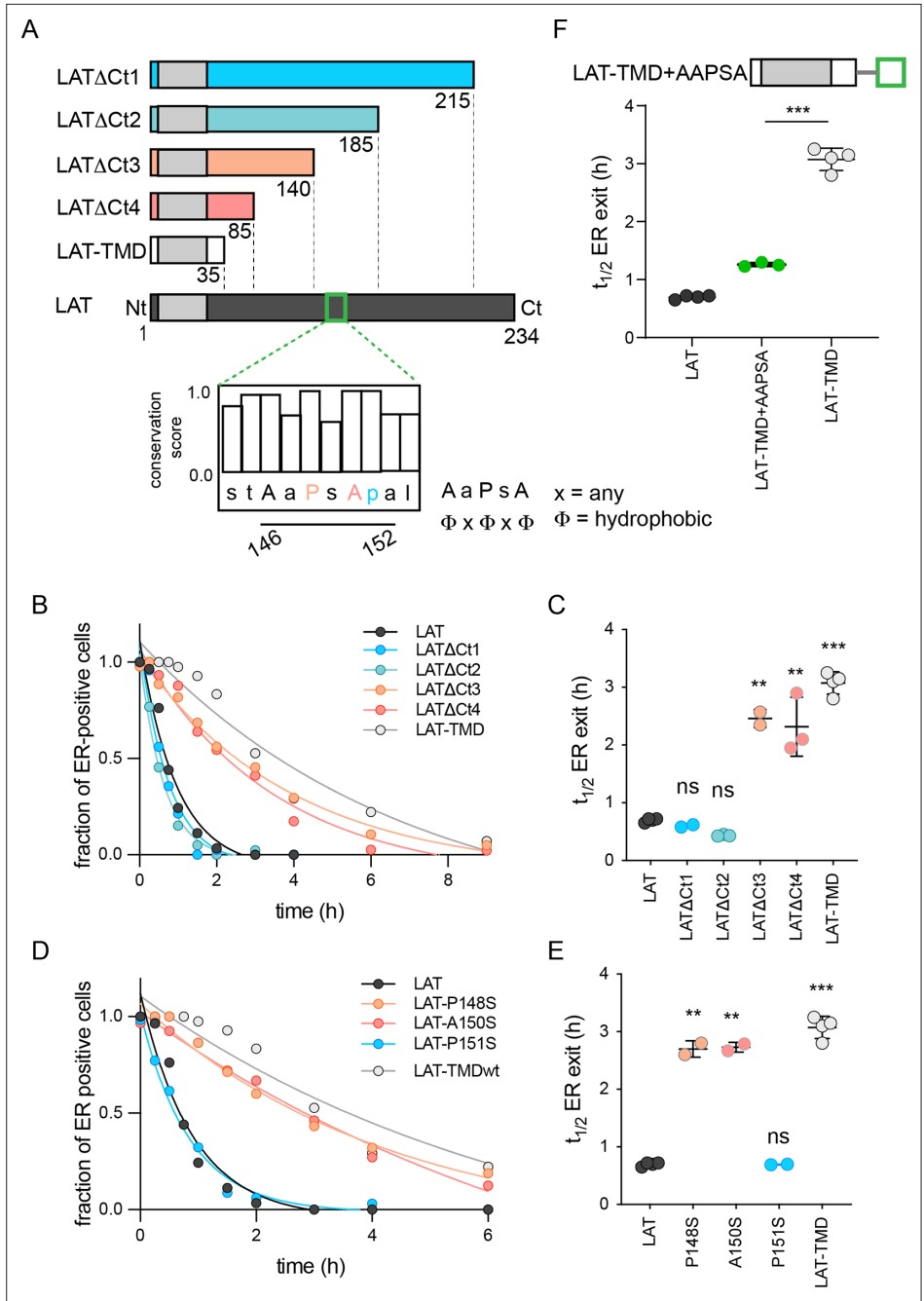

**Figure 2.** Identification of ER exit motif of LAT. (**A**) Schematic of LAT and the truncated versions used here. Inset shows a putative COPII association motif and the evolutionary conservation of residues 144–153 of hLAT. (**B**) Temporal dependence of the fraction of ER-positive cells for LAT truncations. Deletion of a region comprised of residues 140–185 leads to slow ER exit. (**C**) Fitted ER exit kinetics for the constructs represented in panel **C**. Deletion of amino acids 140–185 slows ER exit kinetics by ~fourfold. (**D**) Temporal dependence of fraction of ER-positive cells with point mutations of ΦxΦxΦ motif. Mutations of key residues within the motif slow ER exit kinetics. (**E**) Fitted ER exit kinetics for point mutants in panel **D**. (**F**) Insertion of AaPsA motif into LAT-TMD accelerates ER exit kinetics. (**B and D**) show a representative experiment with exponential decay fits; points in **C, E, and F** represent $t_{1/2}$ values of ER exit from fits of independent repeats with >20 cells/experiment. **\*\*p<0.01, \*\*\*p<0.001, nsp >0.05. Original data quantification can be found in the Source Data files.

The online version of this article includes the following source data and figure supplement(s) for figure 2:

**Source data 1.** Quantification of the temporal dependence of the fraction of ER-positive cells on cytosolic domains.

*Figure 2 continued on next page*

*Figure 2 continued*

**Source data 2.** Quantification of the ER exit kinetics for the constructs represented in panel B.

**Source data 3.** Quantification of the temporal dependence of the fraction of ER-positive cells point mutations in the potential COP-II recognition motif of the cytoslic tail of LAT.

**Source data 4.** Quantification of the fit ER exit kinetics for point mutants in LAT protein show that changes in a single amino acid of the COP-II recognition motif slow down the full-length ER exit to the level of the TMD-only.

**Figure supplement 1.** COPII binding motif mediates fast ER exit of LAT.

---

nearly identically to the full-length (*Figure 3B–C*). LAT-TMD, which has almost no residues that could interact with cytosolic trafficking machinery, had similar Golgi-exit kinetics as full length VSVG (*Figure 3—figure supplement 1*), a model transmembrane secretory cargo. The timescale for efflux of VSVG from the Golgi ($t_{1/2} \sim 0.5$ hr) measured here with RUSH is consistent with a previous measurement by an orthogonal method relying on temperature-sensitive transport (*Hirschberg et al., 1998*).

To validate these conclusions in a different experimental paradigm, we measured Golgi exit kinetics after a temperature block. It was previously shown that culturing mammalian cells at 15–20°C leads to accumulation of secretory cargo in the Golgi (*Milgram and Mains, 1994*; *Venditti et al., 2019*). To accumulate our probes in the Golgi, we used the ER-RUSH constructs to initially restrict them to the ER (*Figure 3D*, left). Biotin was then added at 17 °C for 3 hr to release the protein from the ER and allow sufficient time to accumulate in the Golgi. The cells were then shifted to 37 °C to synchronize Golgi export. As with Golgi-RUSH, LAT largely exited the Golgi within 90 min, whereas LAT-allL was significantly slower, with notable Golgi accumulation remaining after 2 hr (*Figure 3D*). The kinetics of the temperature-block experiment were similar to Golgi RUSH, with raft-preferring LAT exiting Golgi ~twofold faster than non-raft LAT-allL. Based on these observations, we conclude that TMD-encoded raft affinity is an essential determinant of Golgi exit kinetics for LAT.

## Kinetic model of secretory traffic

We attempted to rationalize our observations of ER and Golgi efflux using a kinetic model wherein the secretory kinetics of a TMP are determined by equilibrium partitioning between coexisting compartments in the ER and Golgi (*Figure 4A*). In the model, partitioning to an ER-exit compartment (ERex; analogous to a cellular ER exit site) allows ER efflux with first-order rate constant $k_a$. Full-length and TMD-only proteins have different partition coefficients into this ERex compartment, represented by two free fit parameters $K_{p,ERex\_full}$ and $K_{p,ERex\_TMD}$.

Analogously in the Golgi, we model coexisting compartments that enrich raft or non-raft proteins, again with partition coefficients (i.e. $K_{p,raft\_LAT}$ and $K_{p,raft\_allL}$) describing the ratio of a protein's concentration in the two compartments. These values were set by measurements of $K_{p,raft}$ for the various constructs in GPMVs (*Figure 1A–B*), with the assumption that raft affinities measured at the PM are representative of those in the Golgi. A first-order rate constant $k_b$ is used to describe the transfer of proteins from the Golgi raft sub-compartment to post-Golgi compartments. This conceptual model can be represented via a series of coupled differential equations describing the temporal evolution of abundance of various constructs in three compartments (ER, Golgi, post-Golgi). We emphasize that this model is highly simplified and does not include many features relevant to trafficking of endogenous proteins. Rather, the goal was to identify the minimal set of features that could describe the secretory behavior of our defined set of probes.

This simple model (four free parameters) was used to simultaneously fit experimental observations of eight independent data sets, i.e. release from the ER or from the Golgi for four different protein constructs (analogous to experiments described in *Figures 1 and 3*, respectively). For comparison to the models, experimental datasets were re-measured via live-cell imaging (see *Figure 4—figure supplement 1*) to directly quantify relative construct abundance in the ER or Golgi. Model global fits nicely reproduced the general experimental features of all constructs (*Figure 4B*), though with notable divergences. The general shape of both ER and Golgi efflux curves were well described by the model and the specific Golgi efflux kinetics for all four proteins were modeled quite accurately (*Figure 4B*, filled symbols). Consistent with expectation, the fitted ER exit partition coefficient ($K_{p,ERex\_full}$) for full-length constructs (i.e. those containing the putative COPII binding motif) was ~20-fold greater than for TMD-only constructs. In contrast, the specific ER release kinetics could not be completely reproduced,

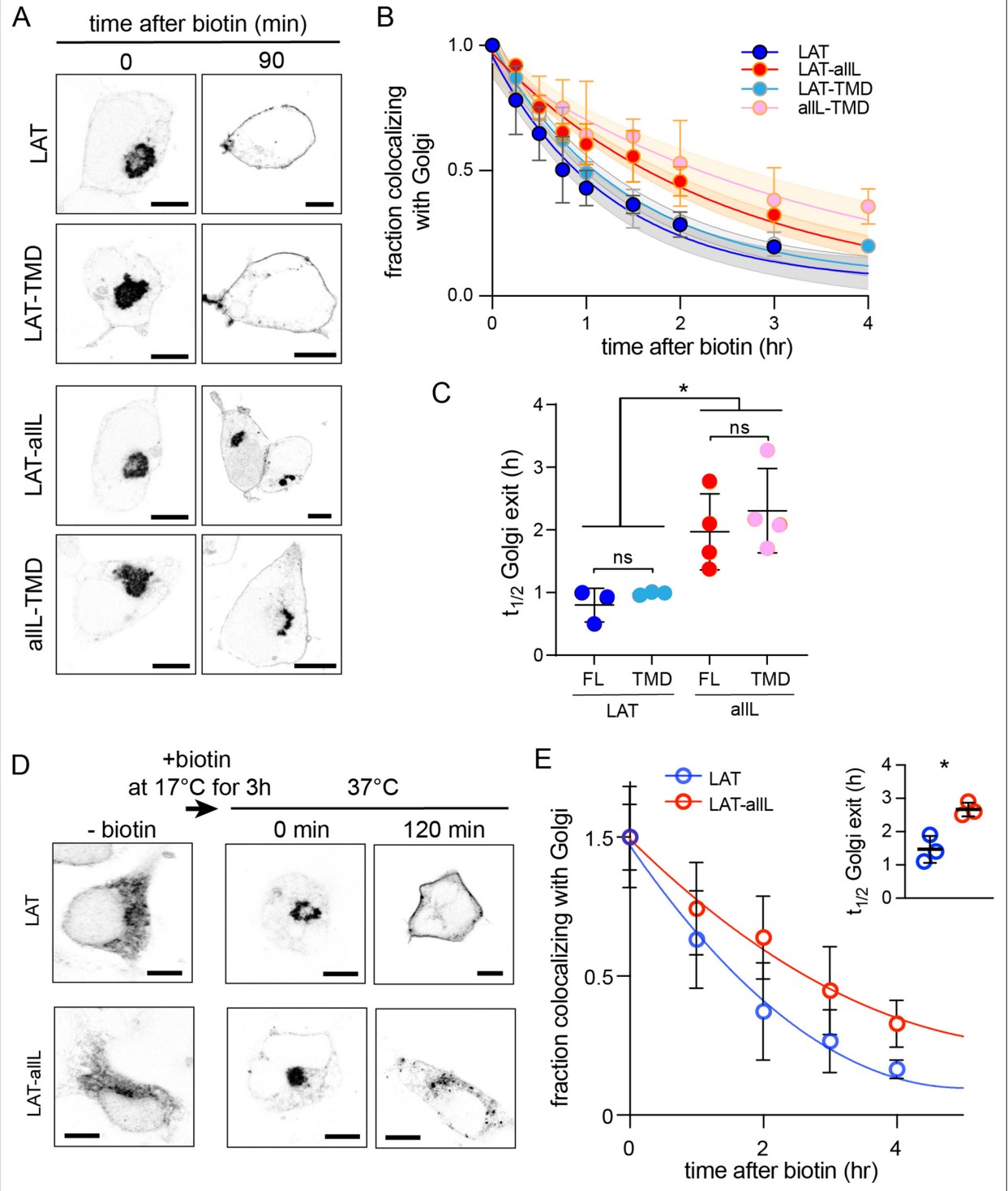

**Figure 3.** Golgi exit kinetics of LAT are dependent on its association with raft domains. (**A**) Representative confocal images of Golgi RUSH experiment show notable Golgi retention of non-raft constructs (LAT-allL and allL-TMD) after 90 min of biotin addition, in contrast to raft-preferring LAT and LAT-TMD. (**B**) Temporal reduction of protein constructs remaining in Golgi after biotin addition (i.e. release from Golgi RUSH), quantified by immunostaining and colocalization with Golgi marker (Giantin, see *Figure 3—figure supplement 2*). Symbols represent average +/-st.dev. from three independent experiments with >15 cells/experiment. Fits represent exponential decays; shading represents 95% confidence intervals. (**C**) Golgi exit rates for raft-associated LAT constructs are ~2.5-fold faster than non-raft versions for both full-length and TMD-only. Points represent $t_{1/2}$ values from fits of independent repeats with >20 cells/experiment. *p<0.05, ns p >0.05. (**D**) Representative confocal images of full-length LAT and LAT-allL during Golgi temperature block. Addition of biotin at 17 °C releases ER-RUSH constructs but traps them in Golgi. Removing temperature block by incubation at

*Figure 3 continued on next page*

*Figure 3 continued*

37 °C leads to fast PM trafficking of LAT but not non-raft LAT-allL. (**E**) Fraction of proteins in Golgi for the constructs shown in D, calculated as in B. Fits represent exponential decays. (inset) Golgi exit kinetics quantified as in C. All scale bars = 5 µm. Original data quantification can be found in the Source Data files.

The online version of this article includes the following source data and figure supplement(s) for figure 3:

**Source data 1.** Quantification of the temporal reduction of protein constructs remaining in Golgi after biotin addition.

**Source data 2.** Quantification of Golgi exit rates.

**Source data 3.** Quantification of the time residency of constructs in Golgi calculated from the kinetics.

**Source data 4.** Quantification of the temporal kinetics after temperature block in the Golgi.

**Figure supplement 1.** Fraction of RUSH-VSVG in Golgi after biotin addition.

**Figure supplement 2.** Example of quantification of Golgi residence for protein of interest (POI).

with the model underestimating ER efflux rate for both constructs with raft-preferring TMDs, and overestimating those of the non-raft proteins (i.e. allL) (*Figure 4B*, open symbols).

These fits could not be improved by adding a Golgi-to-ER retrieval path, consistent with none of our constructs possessing a known retrieval motif (i.e. KDEL) (not shown). However, excellent agreement between the model and all experimental observations could be obtained by allowing the ERex partition coefficients of LAT and LAT-allL to independently vary (*Figure 4C*). Put another way, in this 5-parameter fit, the TMD of these constructs was allowed to influence their affinity for the ER exit compartment. The best-fit was obtained when LAT with a wild-type TMD had ~sixfold greater affinity for the ERex than LAT-allL.

The general agreements between the model and observations support the plausibility of an underlying hypothesis that partitioning between coexisting membrane domains could explain the inter-organelle transfer kinetics in our study. A key aspect of the model is that selective Golgi membrane domains are important for Golgi exit kinetics of raft-associated cargo. To test this inference, we used inhibitors to block the production of raft-forming lipids (*Lasserre et al., 2008*; *Leventhal et al., 2017*) and tested their effects on trafficking rates of our probes. Specifically, sphingolipid and cholesterol synthesis were inhibited by treatment with 25 µM myriocin and 5 µM Zaragozic acid (MZA), respectively, for 2 days. This treatment was previously shown by us and others to reduce ordered membrane domain stability (*Lasserre et al., 2008*; *Leventhal et al., 2017*; *Wang et al., 2023*). We observed significantly reduced Golgi exit kinetics of LAT-TMD in MZA-treated cells, evidenced by clear Golgi localization up to 2 hr after release of Golgi-RUSH, in contrast to control cells (*Figure 4D*). Quantifying exit kinetics by imaging Golgi residence, we observed that the half-time for Golgi exit was significantly increased (by ~60%) for the raft-preferring TMD but not for the non-raft allL-TMD (*Figure 4E–F*).

## Segregation of raft from non-raft proteins in Golgi compartments

To directly image separation between raft and non-raft probes in the Golgi, we co-transfected RUSH versions of LAT-TMD and allL-TMD with the Golgin84-hook to reversibly accumulate both constructs in the Golgi. Their colocalization was then imaged using super-resolved Structured Illumination Microscopy (SIM) (*Figure 5A–B*). Prior to introduction of biotin, the two constructs showed near-perfect colocalization, as expected from their association with the same 'hook'. However, 5 min after releasing the proteins with biotin, their colocalization was significantly reduced (*Figure 5C*), with areas of LAT-TMD and allL-TMD enrichment apparent within the general morphology of the Golgi (*Figure 5B*). Similar segregation of LAT-TMD from allL-TMD could be observed when both were accumulated in the Golgi via a 20 °C temperature block (*Figure 5D*). Notably, such segregation was not observed when LAT-TMD was co-accumulated in the Golgi with either the full-length LAT or with raft-preferring GPI-GFP, which colocalized nearly completely with each other (*Figure 5D*).

To support these observations without relying on temperature modulation, we treated cells with a short-chain Ceramide (D-Ceramide-C6, Cer-C6) which was previously reported to disrupt Golgi organization and post-Golgi transport (*Capasso et al., 2017*; *Duran et al., 2012*; *van Galen et al., 2014*). We combined this treatment with the Golgi-RUSH (Golgin84-hook) constructs to determine the fate of raft- and non-raft probes when their Golgi exit is blocked. Trapping both constructs in Golgi via RUSH produced very high co-localization, as expected (*Figure 5F*). When the constructs were released with biotin in cells treated with Cer-C6, LAT-TMD and allL-TMD both remained in a

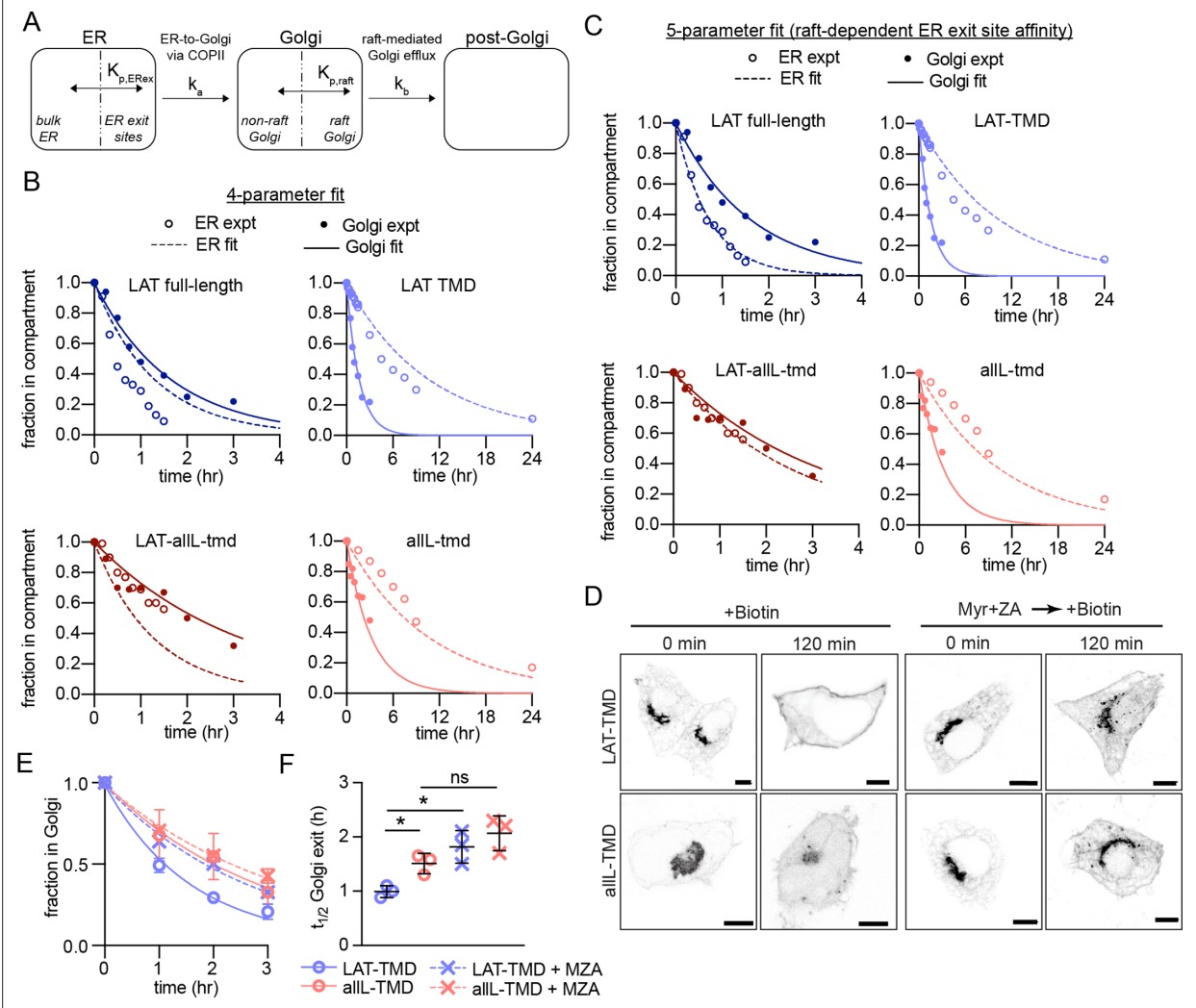

**Figure 4.** Kinetic model describing secretory traffic of LAT-based constructs. (**A**) Schematic of kinetic model. (**B**) Global fit of model with four free parameters to ER and Golgi RUSH data for four experimental constructs. (**C**) Global fit of model with five free parameters (different $K_{p,ERex}$ for LAT and LAT-allL). (**D**) Representative confocal images of Golgi RUSH experiments show notable Golgi retention for raft-preferring LAT after 2 day pre-treatment with Myr +ZA. Scale bars = 5 μm. (**E**) Temporal dependence of the fraction of protein constructs remaining in Golgi after biotin addition (to release from Golgi RUSH). Symbols represent average +/-st dev from three independent experiments with >15 cells/exp. (**F**) Golgi exit rate for the raft-probe LAT-TMD is reduced when raft lipid synthesis is inhibited by Myr-ZA treatment. Points represent $t_{1/2}$ values of Golgi exit from fits of independent repeats with >20 cells/experiment. **p<0.01, [ns]p >0.05. Original data quantification can be found in the Source Data files.

The online version of this article includes the following source data and figure supplement(s) for figure 4:

**Source data 1.** Quantification of the temporal dependence of the fraction of protein constructs remaining in Golgi after biotin addition with and without treatments to abolish raft formation.

**Source data 2.** Quantification of the Golgi exit rate when raft lipid synthesis is inhibited by Myr-ZA treatment.

**Figure supplement 1.** Representative images of the experiments measuring ER exit kinetics within individual cells, used for the kinetic modeling.

perinuclear compartment with the general morphology of the Golgi, but their colocalization was notably reduced, with raft-TMD-rich and -depleted areas visible under confocal imaging (**Figure 5F**). This reduced colocalization was also observed without biotin present, suggesting that Cer-C6 led to a remodeling of the Golgi and associated spatial segregation of raft from non-raft probe proteins, as previously proposed (**van Galen et al., 2014**).

These observations are consistent with previous reports that proteins can segregate in Golgi (**Chen et al., 2017**; **Patterson et al., 2008**). Specifically, TGN46, a resident protein in the vesicular trans-Golgi network (TGN), was shown to physically segregate from a trans-Golgi-resident enzyme

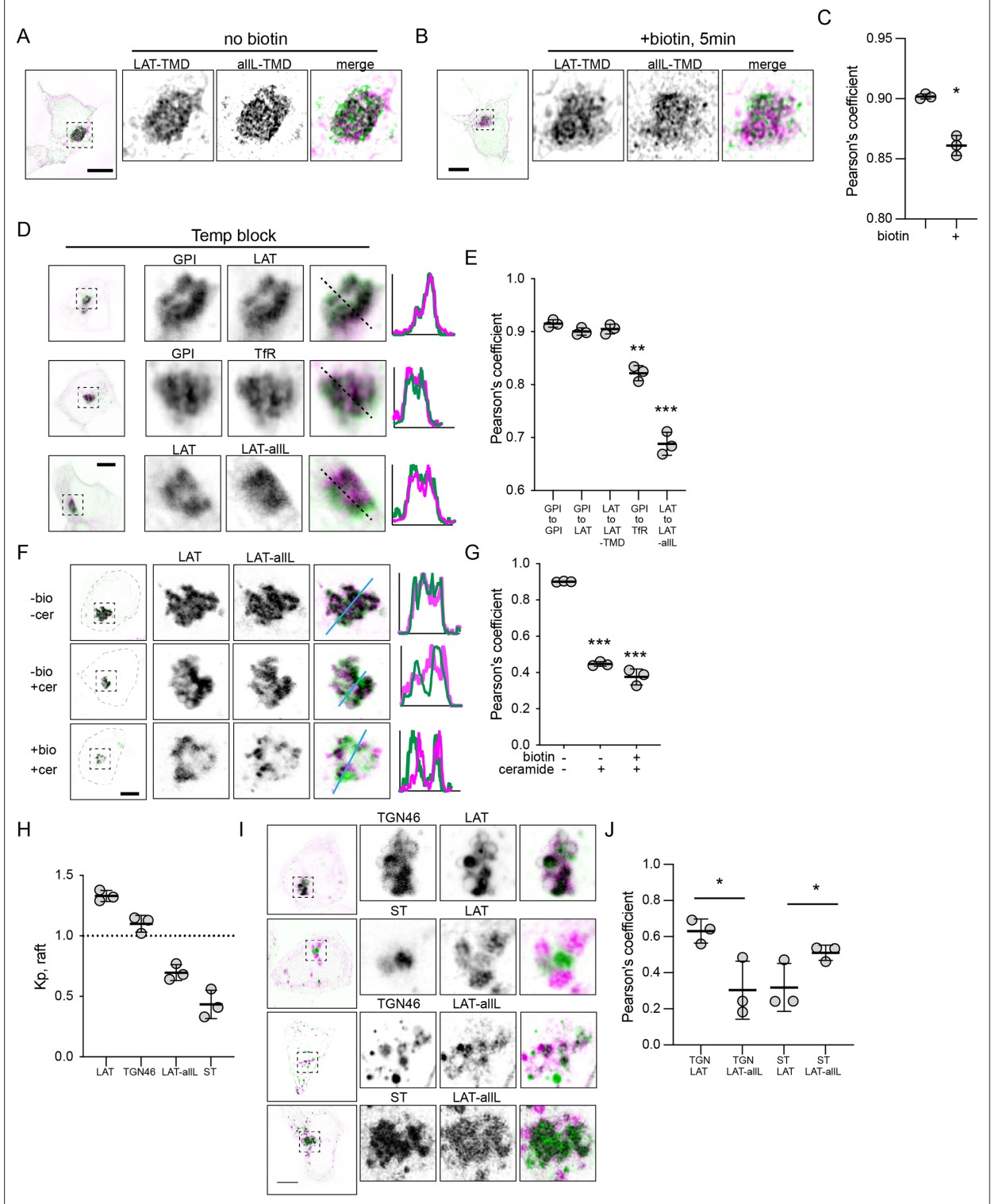

**Figure 5.** Raft probes segregate from non-raft in Golgi. (**A**) Raft vs non-raft probes trapped in Golgi by Golgi-RUSH (i.e. without biotin) and imaged by structured-illumination microscopy (SIM). (**B**) SIM images of localization of raft vs non-raft probes after 5 min of biotin addition to release Golgi RUSH. (**C**) Quantification of colocalization from SIM images by Pearson's coefficient in A and B. Symbols represent average +/-st. dev. from three independent experiments with >15 cells/experiment. (**D**) Confocal images of cellular localization of co-transfected probes in cells grown at 20 °C to accumulate probes in Golgi. (**E**) Quantification of colocalization under the conditions represented in D. Significances shown are relative to LAT-to-LAT-TMD. (**F**) Images of cellular localization of co-transfected Golgi-RUSH probes after treatment with biotin or C6-Cer. (**G**) Quantification of colocalization under the conditions represented in F. Significances shown are relative to -biotin/-Cer. (**H**) Quantification of raft affinity ($K_{p,raft}$) of TGN46 and ST,

*Figure 5 continued on next page*

*Figure 5 continued*

representatives of different Golgi sub-compartments. (**I**) Representative images of raft probes relative to Golgi sub-compartment markers under C6-Cer treatment. (**J**) Quantification of colocalization of proteins represented in I. Symbols in all quantifications are average +/-st. dev. from three independent experiments with >15 cells/experiment. *p<0.05. Original data quantification can be found in the Source Data files.

The online version of this article includes the following source data for figure 5:

**Source data 1.** Quantification of colocalization from SIM images by Pearson's coefficient of raft and non-raft versions in absence or presence of biotin.

**Source data 2.** Quantification of colocalization of several proteins after releasing them from the ER and provoke their block in the Golgi by temperature.

**Source data 3.** Quantification of the Pearson's coefficient after biotin treatment.

**Source data 4.** Quantification of the raft partition coefficients of LAT, TGN46, LAT-allL, and ST.

**Source data 5.** Quantification of co-localizations.

Sialyltransferase (ST) under Cer-C6 treatment (***van Galen et al., 2014***). To determine whether this physical segregation was related to the behavior of our raft domain probes, we co-transfected these Golgi-resident proteins with our probes and imaged them under Cer-C6 treatment. Intriguingly, in Cer-C6-treated cells, the raft probe (LAT-TMD) colocalized well with TGN46 but not with ST (***Figure 5G–H***). Conversely, the non-raft-preferring probe (allL-TMD) colocalized with ST, but not TGN46. These colocalization were consistent with the raft affinity of the Golgi markers evaluated in GPMVs: TGN46 was enriched in the raft phase approximately at parity with LAT-TMD, while ST was depleted from the raft phase (like allL-TMD). These observations support the hypothesis that proteins can segregate in Golgi based on their affinity for distinct membrane domains; however, it is important to emphasize that this segregation does not necessarily imply lateral lipid-driven domains *within* a Golgi cisterna. Reasonable alternative possibilities include separation between cisternae (rather than within), cargo vesicles moving between cisternae, or lateral domains that are mediated by protein assemblies rather than lipids.

## Discussion

The central paradigm of membrane protein traffic involves recognition of cargo sorting motifs by adaptor proteins, which then associate with coat-forming machinery to form inter-organelle trafficking intermediates with compositions that are distinct from their source organelles. Our study suggests that lipid-mediated organization also plays a supporting role in this process.

Several short motifs in cargo cytosolic domains have been reported to mediate recognition by the COPII machinery for ER efflux. Originally identified were di-acidic (e.g. DxE) and di-Leu motifs (***Barlowe, 2003a***), and more recently other sequences with similar roles were identified (***Barlowe, 2003a***; ***Mikros and Diallinas, 2019***; ***Otsu et al., 2013***). Our study reveals that a $\Phi x \Phi x \Phi$ motif mediates fast ER exit of the transmembrane scaffold LAT (***Figure 2***). Notably, constructs missing this motif (including LAT-TMD) still eventually fully exit the ER and reach their steady-state localization, consistent with passive 'leakage' from the ER even in the absence of viable COPII interactions.

Although this motif plays a dominant role in determining LAT ER exit kinetics, our observations suggest a minor contribution of lipid interactions with the protein's TMD. This contribution is revealed in different ER exit kinetics of raft- and nonraft-preferring versions of either full-length or TMD-only constructs (***Figure 1G***). This inference was supported by the kinetic model, which showed the best global fit to experiments when raft-preferring LAT-TMD had a somewhat higher partitioning to ER exit compartments than nonraft LAT-allL (***Figure 4C***). A possible contribution of lipid domains to ER traffic is surprising, as ER cholesterol abundance is believed to be too low to allow formation of ordered lipid domains (***van Meer et al., 2008***). However, synthetic lipid experiments show that cholesterol concentrations as low as 10 mol% can support liquid-ordered phases (***Veatch and Keller, 2002***; ***Veatch and Keller, 2003***) and locally high cholesterol concentrations are possible through localized production, recruitment by proteins, and/or diffusion barriers (***Prasad et al., 2020***). Indeed, enrichment of fluorescent cholesterol at some ER exit sites was recently reported (***Weigel et al., 2021***). We emphasize that our study provides no direct evidence for rafts in the ER and much more extensive characterization would be required to make a convincing claim. However, both our data and model reveal that preferences of certain proteins for ordered membrane regions affect their ER exit rates.

Sorting of protein and lipid cargo in the late secretory pathway is the originally proposed role for raft domains in cells (*Simons and Ikonen, 1997*), supported by the Golgi's relatively high levels of cholesterol, sphingomyelin, and glycolipids (*Jackson et al., 2016*; *van Meer et al., 2008*). Segregation of cargo prior to Golgi exit has been microscopically observed (*Chen et al., 2017*) and anterograde sorting to the PM is facilitated by palmitoylation (*Chum et al., 2016*; *Ernst et al., 2018*), a post-translational modification that imparts raft affinity to many TMPs (*Leventhal et al., 2010*; *Lorent et al., 2017*). PM-directed vesicles in yeast are enriched in sterols and sphingolipids relative to the Golgi, supporting a raft-based sorting model (*Klemm et al., 2009*; *Surma et al., 2011*). Similarly, in mammalian cells, sphingomyelin is enriched in certain Golgi-to-PM carriers (*Deng et al., 2016*; *Sundberg et al., 2019*). Our observations are consistent with these results and implicate ordered lipid domains in trafficking of LAT from the Golgi. This conclusion is also consistent with a previously proposed model of transport through the Golgi via rapid partitioning of cargo between two coexisting membrane phases (*Patterson et al., 2008*), with the 'phases' in our model representing coexisting ordered and disordered lipid-driven membrane domains.

Full-length LAT and its isolated TMD have similar Golgi efflux kinetics, both of which are approximately 2-fold faster than nonraft analogs (*Figure 3*). This behavior was observed with either RUSH or temperature-block and supported by our kinetic model, which predicts raft-dependent exit from the Golgi (*Figure 4*). Whether these observations reflect faster movement between cisternae or faster exit at the TGN remains an open question.

Direct visualization of lipid-driven domains remains a challenge, particularly acute in intracellular membranes which are often small and have complex morphologies. Our SIM imaging suggests segregation of raft from nonraft cargo in the Golgi shortly (5 min) after RUSH release (*Figure 5B*), but at this level of resolution, we can only report reduced colocalization, not intra-Golgi protein distributions. Moreover, segregation within a Golgi cisterna would be very difficult to distinguish from cargo moving between cisternae at different rates or exiting via Golgi-proximal vesicles. Perhaps the most striking instance of separation of raft- from nonraft-Golgi is observable after treatment with a short-chain ceramide (C6-cer), which has been reported to disrupt formation of post-Golgi vesicles (*Campelo et al., 2017*; *van Galen et al., 2014*). In these distorted Golgi, we observe selective colocalization of the raft probe with TGN-46, and vice versa for nonraft and ST (*Figure 5*).

Collectively, these observations suggest that raft domains play a major role in Golgi-to-PM traffic for certain cargoes and that raft affinity is the dominant determinant of Golgi efflux kinetics for LAT. This protein belongs to a family of transmembrane adaptor proteins (TRAPs) with similar general structures and functions (*Chum et al., 2016*; *Park and Yun, 2009*), suggesting that these observations may be relevant for this class of proteins and perhaps others without motifs for clathrin- and adaptor-mediated sorting. In contrast, rafts may play a more subordinate role in ER exit, perhaps facilitating the sorting of certain cargo proteins and lipids to ER exit sites.

## Materials and methods

### Cell culture

HEK-293NT (HEK) and HeLa cells were purchased from ATCC and cultured in medium containing 89% Eagle's Minimum Essential Medium (EMEM), 10% FCS, and 1% penicillin/streptomycin at 37 °C in humidified 5% $CO_2$. COS-7 cells were cultured in medium containing the same formulation but with Dulbecco Modified Essential Medium (DMEM) instead of EMEM and under the same conditions. Transfection was done by Lipofectamine 3000 using the protocols provided with the reagents. Four to 6 hr after transfection, cells were washed with PBS and then incubated with serum-free medium overnight. To synchronize the cells, 1 hr before biotin addition, the cells were given full-serum medium. Lipid synthesis inhibitors (25 µM Myriocin and 5 µM Zaragozic Acid) were added for to the medium and given to the cells 2 days before transfection and during the whole experiment. D-Ceramide-C6 was added to the medium alone or in combination with biotin and given to the cells for 4 hr, and then cells were fixed.

### Plasmids and mutations

All single pass protein constructs from the ER were based on the bicistronic GPI RUSH backbone previously described. We replaced the protein and fluorophore by the amino acid sequence of LAT-TMD,

which is NH2-MEEAILVPCVLGLLLLPILAMLMALCVHCHRLP followed by a short linker (GSGS) and monomeric RFP (mRFP). A full length (LAT, LATallL, LAX) and TMD library (all-TMD, allA8L-TMD, LAT-TMD6Dendo, TfR-TMD, LAX-TMD, VSVG-TMD, LATDC1-4, P148E, P148S, P148A, A150S, P151S, P151A, A183S) were generated by amplifying the sequence of interest by PCR and subsequent cloning of the mutant sequence into the LAT-TMD RUSH construct with KDEL hook to synchronize from the ER. Several constructs were purchased from Addgene: full-length RUSH version of VSVG (#65300). All the single pass RUSH constructs from Golgi were based on the bicistronic VSVG RUSH backbone kindly donated by Jennifer Lippincott-Schwartz lab. We cut the construct with BamHI and EcoRI and replaced the protein with LAT, LATallL, LAT-TMD or all-TMD by amplifying the sequence of interest by PCR. TGN46 and ST plasmids were provided by Felix Campelo lab.

## Imaging

Unless specified, imaging was performed on a confocal microscope using appropriate filters for GFP/RFP fluorescence for transfected plasmids. Single-cell tracking live imaging was performed using u-dish Grid-50 glass bottom plates (Ibidi GmbH) to relocate the selected cells at each acquisition after incubating at 37 °C in the incubator when not imaging. When needed, cells were fixed using 4% paraformaldehyde (PFA), 10 min at RT. Anti-Giantin (1:1000 Rabbit polyclonal, Abcam ab80864) was used to create Golgi mask.

## RUSH expression and chase

In brief, transfected cells with different plasmids containing the RUSH retention system were incubated overnight at 37 °C, and then incubated in the presence of 100 µM biotin (Sigma-Aldrich) to release the cargo proteins. Cells were incubated at 37 °C for various chase times and either directly imaged or fixed with PFA as described above.

## GPMVs and Kp quants

Cell membranes were stained with 5 µg/ml of FAST-DiO (Invitrogen), green fluorescent lipid dye that strongly partitions to disordered phases (*Levental and Levental, 2015a*). Following staining, GPMVs were isolated from transfected HEK-293, or HeLa cells as described (*Sezgin et al., 2012*)(cell type had no effect on results). Briefly, GPMV formation was induced by 2 mM N-ethylmaleimide (NEM) in hypotonic buffer containing 100 mM NaCl, 10 mM HEPES, and 2 mM CaCl2, pH 7.4. To quantify protein partitioning, GPMVs were observed on an inverted epifluorescence microscope (Nikon) at 4 °C after treatment with 200 µM DCA to stabilize phase separation; this treatment has been previously demonstrated not to affect raft affinity of various proteins (*Zhou et al., 2013*). The partition coefficient ($K_{p,raft}$) for each protein construct was calculated from fluorescence intensity of the construct in the raft and non-raft phase for >10 vesicles/trial (e.g. *Figure 1*), with multiple independent experiments for each construct.

## Conservation analysis

After discarding uncharacterized proteins and taking just the first 100 results from 246 hits in the Blast for LAT sequence extracted from Uniprot database, we have used Unipro Ugene to analyze the results of 30 species with >80% similarity.

## Kinetic model for trafficking of RUSH constructs

Residence fraction data were analyzed globally by numerically solving a first-order, homogeneous system of equations that accounts for intra- and inter-compartment transport. A schematic of the trafficking model is shown in *Figure 4A*. Briefly, the ER was modeled with two sub-compartments representing bulk and exit sites, with an intra-ER partition coefficient $K_{p,ERex}$, defined as the ratio of concentrations between the exit and bulk compartments. Similarly, the Golgi was modeled with two sub-compartments representing raft and non-raft sites, with an intra-Golgi partition coefficient $K_{p,raft}$, that is the ratio of concentrations between the raft and nonraft compartments. These values are set in the model via measurements of $K_{p,raft}$ in GPMVs (*Figure 1*; *Diaz-Rohrer et al., 2023*; *Lorent et al., 2017*). Inter-compartment trafficking from ER to Golgi initiates from ER exit sites and terminates at the Golgi with a rate constant $k_a$, while unidirectional post-Golgi traffic originating from Golgi raft sites

was modeled with a rate constant $k_b$. In total, the trafficking model for a given construct is a function of two rate constants and two partition coefficients, one of which is fixed experimentally.

Eight data sets representing the ER and Golgi efflux kinetics of LAT, LAT-TMD, AllL-LAT, and AllL-TMD were fit to the trafficking model. We performed a single global analysis (i.e. simultaneous fit of all eight data sets) using $k_a$, $k_b$, and two $K_{p,ERex}$ (one for full-length and one for TMD-only) as free fit parameters. A second analysis included a third $K_{p,ERex}$, allowing LAT and LAT-allL to be different.

All analysis was performed with custom code written in Mathematica v.12.2 (Wolfram Research Inc, Champaign, IL). Model parameters were optimized with a Levenberg-Marquardt algorithm implemented in the built-in Mathematica function NonlinearModelFit. The target function for minimization was the sum of squared residuals for the combined data sets, $\chi^2 = \sum_i \sum_j \left( y_{ij} - \bar{y}_{ij} \right)^2$, where $i$ indexes the eight data sets, $j$ indexes the data points within a data set, $y_{ij}$ is the observed residence fraction, and $\bar{y}_{ij}$ is the predicted residence fraction. To improve the probability of finding the global minimum, the optimization was repeated $10^3$ times with different random initial values for the adjustable parameters; the best-fit parameters associated with the overall lowest $\chi^2$ value is reported as the solution. The code for the model is available at: https://zenodo.org/doi/10.5281/zenodo.10478607.

## Acknowledgements

We thank members of the Levental lab for their discussions and critical feedback. Funding for this work was provided by the NIH/National Institute of General Medical Sciences (GM124072, GM134949, F32GM145028 to CRS), the Volkswagen Foundation (grant 93091), and the Human Frontiers Science Program (RGP0059/2019). Flow cytometry was performed in the University of Virginia Flow Cytometry Core, RRID: SCR_017829. All authors have no competing interests. The order of equally contributing authors is arbitrary. PL and FC acknowledge support from the Government of Spain (RYC-2017–22227, PID2019-106232RB-I00/10.13039/501100011033; Severo Ochoa CEX2019-000910-S), Fundació Cellex, Fundació Mir-Puig, and Generalitat de Catalunya (CERCA, AGAUR).

## Additional information

### Competing interests

Felix Campelo: Senior Editor, eLife. The other authors declare that no competing interests exist.

### Funding

| Funder | Grant reference number | Author |
| --- | --- | --- |
| National Institute of General Medical Sciences | GM124072 | Ilya Leventall |
| National Institute of General Medical Sciences | GM134949 | Ilya Leventall |
| National Institute of General Medical Sciences | F32GM145028 | Carolyn R Shurer |
| Volkswagen Foundation | 93091 | Kandice R Leventall Ilya Leventall |
| Human Frontier Science Program | RGP0059/2019 | Ilya Leventall |
| Ministerio de Economía y Competitividad | RYC-2017-22227 | Pablo Lujan Felix Campelo |
| Agencia Estatal de Investigación | PID2019-106232RB-I00 | Pablo Lujan Felix Campelo |
| Government of Spain | Severo Ochoa Program CEX2019-000910-S | Pablo Lujan Felix Campelo |
| Fundació Cellex | | Pablo Lujan Felix Campelo |

| Funder | Grant reference number | Author |
|---|---|---|
| Fundació Mir-Puig | | Pablo Lujan<br>Felix Campelo |
| Generalitat de Catalunya | | Pablo Lujan<br>Felix Campelo |

The funders had no role in study design, data collection and interpretation, or the decision to submit the work for publication.

## Author contributions

Ivan Castello-Serrano, Conceptualization, Data curation, Formal analysis, Validation, Investigation, Visualization, Methodology, Writing - original draft; Frederick A Heberle, Formal analysis, Investigation, Methodology; Barbara Diaz-Rohrer, Conceptualization, Investigation; Rossana Ippolito, Data curation, Methodology; Carolyn R Shurer, Data curation, Investigation; Pablo Lujan, Investigation, Methodology; Felix Campelo, Supervision, Methodology; Kandice R Levental, Supervision, Project administration, Writing - review and editing; Ilya Levental, Supervision, Funding acquisition, Project administration, Writing - review and editing

## Author ORCIDs

Ivan Castello-Serrano http://orcid.org/0000-0002-2307-6476
Felix Campelo http://orcid.org/0000-0002-0786-9548
Kandice R Levental http://orcid.org/0000-0002-2234-3683
Ilya Levental http://orcid.org/0000-0002-1206-9545

Joint public review: https://doi.org/10.7554/eLife.89306.3.sa1
Author response https://doi.org/10.7554/eLife.89306.3.sa2

# Additional files

## Supplementary files

- Supplementary file 1. Raft affinity ($K_{p,raft}$) values for constructs used in this study.
- MDAR checklist

## Data availability

Source data files have been provided containing the numberical data to generate the graphs and figures. The kinetic model code can be found here: https://zenodo.org/records/10478608.

The following dataset was generated:

| Author(s) | Year | Dataset title | Dataset URL | Database and Identifier |
|---|---|---|---|---|
| Heberle F | 2024 | Kinetic model for trafficking of RUSH constructs | https://doi.org/10.5281/zenodo.10478608 | Zenodo, 10.5281/zenodo.10478608 |

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
