## [Editor Report · eLife assessment]

In this **important** study, Castello-Serrano and colleagues describe, model and quantify the role of transmembrane domains in protein sorting in the secretory pathway, first at the ER and subsequently at the Golgi. **Convincing** data support the role of a cytoplasmic motif in ER exit, while further experiments are necessary to support a direct connection between the phase partitioning capability of the transmembrane regions and the sorting potential of domains at the Golgi/TGN.

---

## [Referee Report · Joint public review]

In their revised manuscript additional experiments have been conducted competently, and the interpretation of experiments regarding exit from the ER are convincing. They collectively indicate that the phase partitioning behaviour of the TMDs do not have a significant effect on exit from the ER; they all exit the ER very slowly unless they carry a short cytoplasmic domain from LAT which is sufficient to accelerate ER exit. This data is consistent with available literature supporting a role for a ER-exit signal. Along with new experiments in their revision, they have also toned down the assertion that their data rule out a phase partitioning mechanism at the ER.

The authors, however, continue to interpret their experiments regarding Golgi exit of the transmembrane peptides (with luminal and cytoplasmic domains) as conclusive evidence of the role of lipid rafts in exit from the Golgi. This is once again based on correlation of the phase partitioning behaviour of these proteins in GPMVs, phase separated at low temperatures. They argue that this represents very strong evidence that trafficking out of the Golgi is driven by phase separation. The reviewers consider that there are a number of potential issues with the final model that need to be addressed.

We reiterate that:

(1) the phase segregation in the GPMV at low temperatures is dictated by thermodynamics given its composition and the measurement temperature. However at physiological temperatures at which membrane trafficking is taking place these GPMVs will not exhibit phase separation. Hence it is difficult to argue that a sorting mechanism based solely on the partitioning of the synthetic TM constructs into liquid ordered domain detected at low temperatures in GPMVs provide an explanation of the explanation of the differential kinetics of traffic of the LAT TMD and the allL-TMD constructs, although there is a strong correlation with its phase partitioning behaviour.

(2) The fluctuations of lipid composition resembling lo-domains if persisting at higher temperatures and its conversion into a sorting domain will require a cellular mechanism, that may or may not retain similar properties of these lipidic environments. Additionally, TMDs from TfR/VsVG and GPI prefer different domains in the GPMV assays (Table S1) yet they traffic to the cell surface equally rapidly.

(3) The authors fail to discuss the point raised about the relatively high colocalization of TfR with the GPI probe (seen in Fig 5E) in the Golgi. This is inconsistent with their explanation of traffic correlating with partitioning into distinct domains in the Golgi, since TfR and GPI probes show an opposite preference for lo versus ld domains in cooled GPMVs. TMD-allL and the LAT-allL are segregated from TfR in the Golgi, and end up in a different final destination (ie lysosomes). This could represent yet another membrane specialisation in the Golgi for lysosomal traffic. The segregation that the authors report in the Golgi is therefore not a convincing argument for phase preferences in GPMVs dictating the trafficking behaviour of these molecules towards the plasma membrane.

(4) Despite the authors' claim in their rebuttal that 'we feel that GPMVs are a useful tool for quantifying protein preference for ordered (raft) membrane domains and that this preference is a useful proxy for the raft-associated behavior ... biological membrane with a relevant and measurable cellular outcome, specifically inter-organelle trafficking rates." -several caveats for these observations need to be addressed before they constitute strong evidence for the raft model of membrane trafficking proposed. Phase partitioning in GPMVs is just another operational definition and while more refined (ie the data is derived from the membrane of interest, ie, the plasma membrane) it is not very different conceptually from quantitative measurements of detergent-insolubility.

(5) Further work is necessary to establish that ordered domains are formed at the Golgi at physiological temperatures, into which these proteins may partition; subsequently, there must be a mechanism that selectively traffics these proteins towards the cell surface.

(6) The authors continue to conflate thermodynamic phase separation mechanisms with the real possibility of the formation of functional sorting domains by cellular mechanisms that likely involve lipidic interactions, adding to the confusion in the literature.

---

## [Author Response]

The following is the authors’ response to the original reviews.

Editor’s summary:This paper by Castello-Serrano et al. addresses the role of lipid rafts in trafficking in the secretory pathway. By performing carefully controlled experiments with synthetic membrane proteins derived from the transmembrane region of LAT, the authors describe, model and quantify the importance of transmembrane domains in the kinetics of trafficking of a protein through the cell. Their data suggest affinity for ordered domains influences the kinetics of exit from the Golgi. Additional microscopy data suggest that lipid-driven partitioning might segregate Golgi membranes into domains. However, the relationship between the partitioning of the synthetic membrane proteins into ordered domains visualised ex vivo in GPMVs, and the domains in the TGN, remain at best correlative. Additional experiments that relate to the existence and nature of domains at the TGN are necessary to provide a direct connection between the phase partitioning capability of the transmembrane regions of membrane proteins and the sorting potential of this phenomenon.The authors have used the RUSH system to study the traffic of model secretory proteins containing single-passtransmembrane domains that confer defined affinities for liquid ordered (lo) phases in Giant Plasma Membrane derived Vesicles (GPMVs), out of the ER and Golgi. A native protein termed LAT partitioned into these lo-domains, unlike a synthetic model protein termed LAT-allL, which had a substituted transmembrane domain. The authors experiments provide support for the idea that ER exit relies on motifs in the cytosolic tails, but that accelerated Golgi exit is correlated with lo domain partitioning.Additional experiments provided evidence for segregation of Golgi membranes into coexisting lipid-driven domains that potentially concentrate different proteins. Their inference is that lipid rafts play an important role in Golgi exit. While this is an attractive idea, the experiments described in this manuscript do not provide a convincing argument one way or the other. It does however revive the discussion about the relationship between the potential for phase partitioning and its influence on membrane traffic.

We thank the editors and scientific reviewers for thorough evaluation of our manuscript and for positive feedback. While we agree that our experimental findings present a correlation between trafficking rates and raft affinity, in our view, the synthetic, minimal nature of the transmembrane protein constructs in question makes a strong argument for involvement of membrane domains in their trafficking. These constructs have no known sorting determinants and are unlikely to interact directly with trafficking proteins in cells, since they contain almost no extramembrane amino acids. Yet, the LATTMD traffics through Golgi similarly to the full-length LAT protein, but quite different from mutants with lower raft phase affinity. We suggest that these observations can be best rationalized by involvement of raft domains in the trafficking fates and rates of these constructs, providing strong evidence (beyond a simple correlation) for the existence and relevance of such domains.

We have substantially revised the manuscript to address all reviewer comments, including several new experiments and analyses. These revisions have substantially improved the manuscript without changing any of the core conclusions and we are pleased to have this version considered as the “version of record” in eLife.

Below is our point-by-point response to all reviewer comments.

ER exit:The experiments conducted to identify an ER exit motif in the C-terminal domain of LAT are straightforward and convincing. This is also consistent with available literature. The authors should comment on whether the conservation of the putative COPII association motif (detailed in Fig. 2A) is significantly higher than that of other parts of the C-terminal domain.

Thank you for this suggestion, this information has now been included as Supp Fig 2B. While there are other wellconserved residues of the LAT C-terminus, many regions have relatively low conservation. In contrast, the essential residues of the COPII association motif (P148 and A150) are completely conserved across in LAT across all species analyzed.

One cause of concern is that addition of a short cytoplasmic domain from LAT is sufficient to drive ER exit, and in its absence the synthetic constructs are all very slow. However, the argument presented that specific lo phase partitioning behaviour of the TMDs do not have a significant effect on exit from the ER is a little confusing. This is related to the choice of the allL-TMD as the 'non-lo domain' partitioning comparator. Previous data has shown that longer TMDs (23+) promote ER export (eg. Munro 91, Munro 95, Sharpe 2005). The mechanism for this is not, to my knowledge, known. One could postulate that it has something to do with the very subject of this manuscript- lipid phase partitioning. If this is the case, then a TMD length of 22 might be a poor choice of comparison. A TMD 17 Ls' long would be a more appropriate 'non-raft' cargo. It would be interesting to see a couple of experiments with a cargo like this.

The basis for the claim that raft affinity has relatively minor influence on ER exit kinetics, especially in comparison to the effect of the putative COPII interaction motif, is in Fig 1G. We do observe some differences between constructs and they may be related to raft affinity, however we considered these relatively minor compared to the nearly 4-fold increase in ER efflux induced by COPII motifs.

We have modified the wording in the manuscript to avoid the impression that we have ruled out an effect of raft affinity of ER exit.

We believe that our observations are broadly consistent with those of Munro and colleagues. In both their work and ours, long TMDs were able to exit the ER. In our experiments, this was true for several proteins with long TMDs, either as fulllength or as TMD-only versions (see Fig 1G). We intentionally did not measure shorter synthetic TMDs because these would not have been comparable with the raft-preferring variants, which all require relatively long TMDs, as demonstrated in our previous work1,2. Thus, because our manuscript does not make any claims about the influence of TMD length on trafficking, we did not feel that experiments with shorter non-raft constructs would substantively influence our conclusions.

However, to address reviewer interest, we did complete one set of experiments to test the effect of shortening the TMD on ER exit. We truncated the native LAT TMD by removing 6 residues from the C-terminal end of the TMD (LAT-TMDd6aa). This construct exited the ER similarly to all others we measured, revealing that for this set of constructs, short TMDs did not accumulate in the ER. ER exit of the truncated variant was slightly slower than the full-length LAT-TMD, but somewhat faster than the allL-TMD. These effects are consistent with our previous measurements with showed that this shortened construct has slightly lower raft phase partitioning than the LAT-TMD but higher than allL2. While these are interesting observations, a more thorough exploration of the effect of TMD length would be required to make any strong conclusion, so we did not include these data in the final manuscript.

**Author response image 1. sa2fig1:** 

Golgi exit:For the LAT constructs, the kinetics of Golgi exit as shown in Fig. 3B are surprisingly slow. About half of the protein Remains in the Golgi at 1 h after biotin addition. Most secretory cargo proteins would have almost completely exited the Golgi by that time, as illustrated by VSVG in Fig. S3. There is a concern that LAT may have some tendency to linger in the Golgi, presumably due to a factor independent of the transmembrane domain, and therefore cannot be viewed as a good model protein. For kinetic modeling in particular, the existence of such an additional factor would be far from ideal. A valuable control would be to examine the Golgi exit kinetics of at least one additional secretory cargo.

We disagree that LAT is an unusual protein with respect to Golgi efflux kinetics. In our experiments, Golgi efflux of VSVG was similar to full-length LAT (t1/2 ~ 45 min), and both of these were similar to previously reported values3. Especially for the truncated (i.e. TMD) constructs, it is very unlikely that some factor independent of their TMDs affects Golgi exit, as they contain almost no amino acids outside the membrane-embedded TMD.

Practically, it has proven somewhat challenging to produce functional RUSH-Golgi constructs. We attempted the experiment suggested by the reviewer by constructing SBP-tagged versions of several model cargo proteins, but all failed to trap in the Golgi. We speculate that the Golgin84 hook is much more sensitive to the location of the SBP on the cargo, being an integral membrane protein rather than the lumenal KDEL-streptavidin hook. This limitation can likely be overcome by engineering the cargo, but we did not feel that another control cargo protein was essential for the conclusions we presented, thus we did not pursue this direction further.

Comments about the trafficking model(1) In Figure 1E, the export of LAT-TMD from the ER is fitted to a single-exponential fit that the authors say is "well described". This is unclear and there is perhaps something more complex going on. It appears that there is an initial lag phase and then similar kinetics after that - perhaps the authors can comment on this?

This is a good observation. This effect is explainable by the mechanics of the measurement: in Figs 1 and 2, we measure not ‘fraction of protein in ER’ but ‘fraction of cells positive for ER fluorescence’. This is because the very slow ER exit of the TMD-only constructs present a major challenge for live-cell imaging, so ER exit was quantified on a population level, by fixing cells at various time points after biotin addition and quantifying the fraction of cells with observable ER localization (rather than tracking a single cell over time).

For fitting to the kinetic model (which attempts to describe ‘fraction in ER/Golgi’) we re-measured all constructs by livecell imaging (see Supp Fig 5) to directly quantify relative construct abundance in the ER or Golgi. These data did not have the plateau in Fig 1E, suggesting that this is an artifact of counting “ER positive cells” which would be expected to have a longer lag than “fraction of protein in ER”. Notably however, t1/2 measured by both methods was similar, suggesting that the population measurement agrees well with single-cell live imaging.

We have included all these explanations and caveats in the manuscript. We have also changed the wording from “well described” to “reasonably approximated”.

(2) The model for Golgi sorting is also complicated and controversial, and while the authors' intention to not overinterpreting their data in this regard must be respected, this data is in support of the two-phase Golgi export model (Patterson et al PMID:18555781).

The reviewers are correct, our observations and model are consistent with Patterson et al and it was a major oversight that a reference to this foundational work was not included. We have now added a discussion regarding the “two phase model” of Patterson and Lippincott-Schwartz.

Furthermore contrary to the statement in lines 200-202, the kinetics of VSVG exit from the Golgi (Fig. S3) are roughly linear and so are NOT consistent with the previous report by Hirschberg et al.

Regarding kinetics of VSVG, our intention was to claim that the timescale of VSVG efflux from the Golgi was similar to previously reported in Hirschberg, i.e. t1/2 roughly between 30-60 minutes. We have clarified this in the text. Minor differences in the details between our observations and Hirschberg are likely attributable to temperature, as those measurements were done at 32°C for the tsVSVG mutant.

Moreover, the kinetics of LAT export from the Golgi (Fig. 3B) appear quite different, more closely approximating exponential decay of the signal. These points should be described accurately and discussed.

Regarding linear versus exponential fits, we agree that the reality of Golgi sorting and efflux is far more complicated than accounted for by either the phenomenological curve fitting in Figs 1-3 or the modeling in Fig 4. In addition to the possibility of lateral domains within Golgi stacks, there is transport between stacks, retrograde traffic, etc. The fits in Figs1-3 are not intended to model specifics of transport, but rather to be phenomenological descriptors that allowed us to describe efflux kinetics with one parameter (i.e. t1/2). In contrast, the more refined kinetic modeling presented in Figure 4 is designed to test a mechanistic hypothesis (i.e. coexisting membrane domains in Golgi) and describes well the key features of the trafficking data.

Relationship between membrane traffic and domain partitioning:(1) Phase segregation in the GPMV is dictated by thermodynamics given its composition and the measurement temperature (at low temperatures 4degC). However at physiological temperatures (32-37degC) at which membrane trafficking is taking place these GPMVs are not phase separated. Hence it is difficult to argue that a sorting mechanism based solely on the partitioning of the synthetic LAT-TMD constructs into lo domains detected at low temperatures in GPMVs provide a basis (or its lack) for the differential kinetics of traffic of out of the Golgi (or ER). The mechanism in a living cell to form any lipid based sorting platforms naturally requires further elaboration, and by definition cannot resemble the lo domains generated in GPMVs at low temperatures.

We thank the reviewers for bringing up this important point. GPMVs are a useful tool because they allow direct, quantitative measurements of protein partitioning between coexisting ordered and disordered phases in complex, cell-derived membranes. However, we entirely agree, that GPMVs do not fully represent the native organization of the living cell plasma membrane and we have previously discussed some of the relevant differences4,5. Despite these caveats, many studies have supported the cellular relevance of phase separation in GPMVs and the partitioning of proteins to raft domains therein 6-9. Most notably, elegant experiments from several independent labs have shown that fluorescent lipid analogs that partition to Lo domains in GPMVs also show distinct diffusive behaviors in live cells 6,7, strongly suggesting the presence of nanoscopic Lo domains in live cells. Similarly, our recent collaborative work with the lab of Sarah Veatch showed excellent agreement between raft preference in GPMVs and protein organization in living immune cells imaged by super-resolution microscopy10. Further, several labs6,7, including ours11, have reported nice correlations between raft partitioning in GPMVs and detergent resistance, which is a classical (though controversial) assay for raft association.

Based on these points, we feel that GPMVs are a useful tool for quantifying protein preference for ordered (raft) membrane domains and that this preference is a useful proxy for the raft-associated behavior of these probes in living cells. We propose that this approach allows us to overcome a major reason for the historical controversy surrounding the raft field: nonquantitative and unreliable methodologies that prevented consistent definition of which proteins are supposed to be present in lipid rafts and why. Our work directly addresses this limitation by relating quantitative raft affinity measurements in a biological membrane with a relevant and measurable cellular outcome, specifically inter-organelle trafficking rates.

Addressing the point about phase transition temperatures in GPMVs: this is the temperature at which macroscopic domains are observed. Based on physical models of phase separation, it has been proposed that macroscopic phase separation at lower temperatures is consistent sub-microscopic, nanoscale domains at higher temperatures8,12. These smaller domains can potentially be stabilized / functionalized by protein-protein interactions in cells13 that may not be present in GPMVs (e.g.because of lack of ATP).

(2) The lipid compositions of each of these membranes - PM, ER and Golgi are drastically different. Each is likely to phase separate at different phase transition temperatures (if at all). The transition temperature is probably even lower for Golgi and the ER membranes compared to the PM. Hence, if the reported compositions of these compartments are to be taken at face value, the propensity to form phase separated domains at a physiological temperature will be very low. Are ordered domains even formed at the Golgi at physiological temperatures?

It is a good point that the membrane compositions and the resulting physical properties (including any potential phase behavior) will be very different in the PM, ER, and Golgi. Whether ordered domains are present in any of these membranes in living cells remains difficult to directly visualize, especially for non-PM membranes which are not easily accessible by probes, are nanoscopic, and have complex morphologies. However, the fact that raft-preferring probes / proteins share some trafficking characteristics, while very similar non-raft mutants behave differently argues that raft affinity plays a role in subcellular traffic.

(3) The hypothesis of 'lipid rafts' is a very specific idea, related to functional segregation, and the underlying basis for domain formation has been also hotly debated. In this article the authors conflate thermodynamic phase separation mechanisms with the potential formation of functional sorting domains, further adding to the confusion in the literature. To conclude that this segregation is indeed based on lipid environments of varying degrees of lipid order, it would probably be best to look at the heterogeneity of the various membranes directly using probes designed to measure lipid packing, and then look for colocalization of domains of different cargo with these domains.

This is a very good suggestion, and a direction we are currently following. Unfortunately, due to the dynamic nature and small size of putative lateral membrane domains, combined with the interior of a cell being filled with lipophilic environments that overlay each other, directly imaging domains in organellar membranes with lipid packing probes remains extremely difficult with current technology (or at least available to us). We argue that the TMD probes used in this manuscript are a reasonable alternative, as they are fluorescent probes with validated selectivity for membrane compartments with different physical properties.

Ultimately, the features of membrane domains suggested by a variety of techniques – i.e. nanometric, dynamic, relatively similar in composition to the surrounding membrane, potentially diverse/heterogeneous – make them inherently difficult to microscopically visualize. This is one reason why we believe studies like ours, which use a natural model system to directly quantify raft-associated behaviors and relate them to cellular effects (in our case, protein sorting), are a useful direction for this field.

We believe we have been careful in our manuscript to avoid confusing language surrounding lipid rafts, phase separation, etc. Our experiments clearly show that mammalian membranes have the capacity to phase separate, that some proteins preferentially interact with more ordered domains, and that this preference is related to the subcellular trafficking fates and rates of these proteins. We have edited the manuscript to emphasize these claims and avoid the historical controversies and confusions.

(4) In the super-resolution experiments (by SIM- where the enhancement of resolution is around two fold or less compared to optical), the authors are able to discern a segregation of the two types of Golgi-resident cargo that have different preferences for the lo-domains in GPMVs. It should be noted that TMD-allL and the LATallL end up in the late endosome after exit of the Golgi. Previous work from the Bonafacino laboratory (PMID: 28978644) has shown that proteins (such as M6PR) destined to go to the late endosome bud from a different part of the Golgi in vesicular carriers, while those that are destined for the cell surface first (including TfR) bud with tubular vesicular carriers. Thus at the resolution depicted in Fig 5, the segregation seen by the authors could be due to an alternative explanation, that these molecules are present in different areas of the Golgi for reasons different from phase partitioning. The relatively high colocalization of TfR with the GPI probe in Fig 5E is consistent with this explanation. TfR and GPI prefer different domains in the GPMV assays yet they show a high degree of colocalization and also traffic to the cell surface.

This is a good point. Even at microscopic resolutions beyond the optical diffraction limit, we cannot make any strong claims that the segregation we observe is due to lateral lipid domains and not several reasonable alternatives, including separation between cisternae (rather than within), cargo vesicles moving between cisternae, or lateral domains that are mediated by protein assemblies rather than lipids. We have explicitly included this point in the Discussion: “Our SIM imaging suggests segregation of raft from nonraft cargo in the Golgi shortly (5 min) after RUSH release (Fig 5B), but at this level of resolution, we can only report reduced colocalization, not intra-Golgi protein distributions. Moreover, segregation within a Golgi cisterna would be very difficult to distinguish from cargo moving between cisternae at different rates or exiting via Golgi-proximal vesicles.”

We have also added a similar caveat in the Results section of the manuscript: “These observations support the hypothesis that proteins can segregate in Golgi based on their affinity for distinct membrane domains; however, it is important to emphasize that this segregation does not necessarily imply lateral lipid-driven domains within a Golgi cisterna. Reasonable alternative possibilities include separation between cisternae (rather than within), cargo vesicles moving between cisternae, or lateral domains that are mediated by protein assemblies rather than lipids.”

Finally, while probes with allL TMD do eventually end up in late endosomes (consistent with the Bonifacino lab’s findings which we include), they do so while initially transiting the PM2,11.

Minor concerns:(1) Generally, the quantitation is high quality from difficult experimental data. Although a lot appears to be manual, it appears appropriately performed and interpreted. There are some claims that are made based on this quantitation, however, where there are no statistics performed. For example, figure 1B. Any quantitation with an accompanying conclusion should be subject to a statistical test. I think the quality of the model fits- this is particularly important.

We appreciate the thoughtful feedback, the quantifications and fits were not trivial, but we believe important. We have added statistical significance to Figure 1B and others where it was missing.

(2) Modulation of lipid levels in Fig 4E shows a significant change for the trafficking rate for the LAT-TMD construct and a not so significant change for all-TMD construct. However, these data are not convincing and appear to depend on a singular data point that appears to lower the mean value. In general, the experiment with the MZA inhibitor (Fig. 4D-F) is hard to interpret because cells will likely be sick after inhibition of sphingolipid and cholesterol synthesis. Moreover, the difference in effects for LAT-TMD and allL-TMD is marginal.

We disagree with this interpretation. Fig 4E shows the average of three experiments and demonstrates clearly that the inhibitors change the Golgi efflux rate of LAT-TMD but not allL-TMD. This is summarized in the t1/2 quantifications of Fig 4F, which show a statistically significant change for LAT-TMD but not allL-TMD. This is not an effect of a singular data point, but rather the trend across the dataset.

Further, the inhibitor conditions were tuned carefully to avoid cells becoming ‘sick’: at higher concentrations, cells did adopt unusual morphologies and began to detach from the plates. We pursued only lower concentrations, which cells survived for at least 48 hrs and without major morphological changes.

(3) Line 173: 146-AAPSA-152 should read either 146-AAPSA-150 or 146-AAPSAPA-152, depending on what the authors intended.

Thanks for the careful reading, we intended the former and it has been fixed.

(4) What is the actual statistical significance in Fig. 3C and Fig. 3E? There is a single asterisk in each panel of the figure but two asterisks in the legend.

Apologies, a single asterisk representing p<0.05 was intended. It has been fixed.

(5) The code used to calculate the model. is not accessible. It is standard practice to host well-annotated code on Github or similar, and it would be good to have this publicly available.

We have deposited the code on a public repository (doi: 10.5281/zenodo. 10478607) and added a note to the Methods.

(1) Lorent, J. H. et al. Structural determinants and func7onal consequences of protein affinity for membrane ra=s. Nature communica/ons 8, 1219 (2017).PMC5663905

(2) Diaz-Rohrer, B. B., Levental, K. R., Simons, K. & Levental, I. Membrane ra = associa7on is a determinant of plasma membrane localiza7on. Proc Natl Acad Sci U S A 111, 8500-8505 (2014).PMC4060687

(3) Hirschberg, K. et al. Kine7c analysis of secretory protein traffic and characteriza7on of golgi to plasma membrane transport intermediates in living cells. J Cell Biol 143, 1485-1503 (1998).PMC2132993

(4) Levental, K. R. & Levental, I. Giant plasma membrane vesicles: models for understanding membrane organiza7on. Current topics in membranes 75, 25-57 (2015)

(5) Sezgin, E. et al. Elucida7ng membrane structure and protein behavior using giant plasma membrane vesicles. Nat Protoc 7, 1042-1051 (2012)

(6) Komura, N. et al. Ra=-based interac7ons of gangliosides with a GPI-anchored receptor. Nat Chem Biol 12, 402-410 (2016)

(7) Kinoshita, M. et al. Ra=-based sphingomyelin interac7ons revealed by new fluorescent sphingomyelin analogs. J Cell Biol 216, 1183-1204 (2017).PMC5379944

(8) Stone, M. B., Shelby, S. A., Nunez, M. F., Wisser, K. & Veatch, S. L. Protein sor7ng by lipid phase-like domains supports emergent signaling func7on in B lymphocyte plasma membranes. eLife 6 (2017).PMC5373823

(9) Machta, B. B. et al. Condi7ons that Stabilize Membrane Domains Also Antagonize n-Alcohol Anesthesia. Biophys J 111, 537-545 (2016)

(10) Shelby, S. A., Castello-Serrano, I., Wisser, I., Levental, I. & S., V. Membrane phase separa7on drives protein organiza7on at BCR clusters. Nat Chem Biol in press (2023)

(11) Diaz-Rohrer, B. et al. Rab3 mediates a pathway for endocy7c sor7ng and plasma membrane recycling of ordered microdomains Proc Natl Acad Sci U S A 120, e2207461120 (2023)

(12) Veatch, S. L. et al. Cri7cal fluctua7ons in plasma membrane vesicles. ACS Chem Biol 3, 287-293 (2008)

(13) Wang, H. Y. et al. Coupling of protein condensates to ordered lipid domains determines func7onal membrane organiza7on. Science advances 9, eadf6205 (2023).PMC10132753